# Digitalisation of Teaching and Learning as a Tool for Increasing Students' Satisfaction and Educational Efficiency: Using Smart Platforms in EFL

**Sandra Stefanovic [1,\*] and Elena Klochkova [2]**

1    Faculty of Engineering, University of Kragujevac, 34000 Kragujevac, Serbia
2    Institute of Humanities, Peter the Great St. Petersburg Polytechnic University, 195251 St. Petersburg, Russia; samarinne@list.ru
\*    Correspondence: sandra_stef@yahoo.com; Tel.: +381-34-335990

**Abstract:** This manuscript aims to present possibilities for developing mobile and smart platforms and systems in teaching and learning the English language for engineering professionals in different engineering study programs. Foreign language teaching and learning processes are based on traditional methods, while in engineering and technical sciences, teaching and learning processes include different digital platforms. Therefore, the following hypotheses were stated. ($H_1$) It is possible to develop a software solution for mobile platforms that can have a higher level of interactivity, and it may lead to better learning outcomes, especially in the field of adopting engineering vocabulary. ($H_2$) Implementation of the developed solution increases motivation for learning and leads to a higher level of satisfaction with the learning process as a part of the quality of life. ($H_3$) Students who have digital and mobile platforms in the learning process could have higher achievement values. This manuscript presents software application development and its implementation in teaching English as a foreign language for engineering and technical study programs on the bachelor level. Initial results in implementation and satisfaction of end users point to the justification of implementing such solutions.

**Keywords:** students' satisfaction; blended learning; e-learning; m-learning; EFL English as foreign language

## 1. Introduction

The development of technology and digitalisation and especially the development of information and communication technologies have made an important and significant change and shift in all human activities, including training, teaching, learning, and education. These new technologies and accompanied approaches also made things different and more straightforward, improving learners' satisfaction. Improvement of learners' satisfaction contributes to the quality of life as a whole.

By their nature, universities and high education institutions have always been more open and ready to adopt new information and technological solutions and new learning tools and platforms that contribute to the improvement of the learning and teaching process. Digitalisation and usage of information and communication technologies have changed all aspects of the educational process, bringing some benefits and bringing some drawbacks. Initial steps start with the development of digitalised, electronic contacts for teaching and learning, starting from static e-learning materials (such as electronic documents, presentations, and e-books) and leading to dynamic solutions (such as remote or virtual laboratories, intelligent tutoring systems, e-classrooms, or augmented reality solutions for e-learning). On the other hand, the complete concept of learning has been changed, moving from traditional forms through some blended forms (where traditional approaches were mixed with more or less e-learning contents and tools) up to almost pure e-learning concepts. In recent months, facing the COVID 19 pandemic, the shift from

traditional learning to more or less e-learning approaches has been evident. On one hand, it contributes to overcoming existing difficulties, but on the other hand, it will be interesting to evaluate final effects and outcomes in the years that will come.

The evolution of the e-learning concept and digitalisation in education started with developing static electronic materials (which were available to the students using different means, platforms, CDs, external memories, etc.) The evolution continued in developing dynamic interactive solutions, platforms, and systems to support teaching and learning processes (Learning management systems, such as Moodle [1]). Learning management systems enabled the organisation of learning materials, providing several functionalities for class management. The following developments included solutions such as virtual and remotely controlled laboratories [2]. Such a system could use information and communication technologies to control entire real systems or be wholly virtual or developed as augmented learning and training systems (Augmented Learning/Training Systems [3,4]). Lately, there is a trend in using widely presented and available platforms such as mobile platforms and smartphones for different purposes, from e-business to e-learning, contributing to the concept of m-learning. There is a piece of evidence that mobile and smartphones could have a valuable contribution at different educational levels, including education at higher education level [5].

According to the performed analysis, study programs at bachelor level in the field of technical and technological sciences have courses that are dedicated to the learning of the foreign language either as professional or business language focused on communication and communication skills (so-called soft skills) [6]. Generally, foreign language is not the main or core course in engineering curricula and study programs; however, knowledge of foreign languages, especially English, has tangible importance in engineering and technical professions [7].

Learning of a foreign language has importance in enabling processional communication of graduated students and plays an essential role in lifelong learning and modern expert and scientific literature [8]. (It is clear that the English language is the de facto language of engineering and the de facto language of science. Almost all journals listed at WoS and the Clarivate lists are in the English language).

Generally, in engineering study programs, especially in engineering sciences, classes have many students and are not suitable for learning English language skills. Besides, in most study programs, the English language is placed in the first or second year. Having this in mind, it is not easy to provide a sufficient number of places in phono—laboratories or direct contact and conversation between teacher and students. Generally, there is a natural form of teaching and learning where course materials (texts) are distributed in e-form using e-mail, web, or Learning management system. It can be concluded that digital contents and digital tools, information, and communication technologies in foreign language learning at engineering study programs are not on a sufficient level.

According to different authors, digitalisation and usage of mobile platforms and systems in teaching and learning of foreign language, as well as the efficiency of developed and implemented systems, appear as exciting research and practical issue [9–15].

Mobile platforms and smartphones as support in teaching and learning of foreign language have been used for different purposes, where students use different possibilities, services and options of mobile devices: SMS, searching of Internet or e-mail. Some researchers have reported on using applications of mobile devices for improvement of interaction and message exchange through e-mails to teach and learn specific language constructions [16]. Other authors focused their research on possibilities of different language content search options [17] or targeted SMSs [18,19], as well as usage of mobile applications and platforms for learning of foreign language from static dictionaries and solutions or even dynamic, adaptive solutions and systems [20–23].

Several studies point to effectiveness and usefulness and to the number of benefits that mobile platforms and devices could bring to teaching and learning foreign languages [24]. Of course, some changes, drawbacks, and open issues exist in developing and implement-

ing such a solution in learning foreign languages [25]. The main drawback of mobile platforms and applications is that these applications mainly focus on learning language by adopting an isolated vocabulary and not language and vocabulary in its natural context. Additional issues are lack of feedback and lack of adaptation to the needs of the specific user. Having the listed facts in mind, advantages and drawbacks of the existing system, position of foreign language in curriculum of engineering study programs, and the importance of this specific course, the new interactive mobile application was developed as a support for the teaching and learning process. It helps the acquisition of professional English and vocabulary in specific engineering context due to a high level of adaptability to the needs and learning paths of the individual student or user. The developed application was introduced and tested to determine its educational effects and contribution to the learning outcomes and students' satisfaction.

## 2. Research Method

It is clear that the usage of information and communication technology and a complete digitalisation trend is necessary in the educational process and learning and teaching of foreign languages [26], especially in cases where foreign language has been studied as professional language in engineering technical study programs. Equipment for visual and audio communication are channels for information transfer to the end-user, and the role of mobile platforms and smartphones is exact. This combination enables significant support to the acquisition of the vocabulary of a foreign language and substantially impacts the quality of the process, the improvement of learning outcomes, and users' satisfaction [27,28].

This research aims to determine the possibilities and effects of mobile application implementation as support in foreign language learning (professional language) on bachelor studies as a part of engineering and technical and technological sciences curricula. The goal is to compare and contrast the effects of usage of such system and students' opinions and satisfaction with this blended approach in learning, utilizing mobile and smart phones in the learning process.

According to the stated goals, the following hypotheses were set up:

**(H 1).** *It is possible to develop a software solution for mobile platforms which will lead to a higher level of interaction and better learning outcomes, especially vocabulary acquisition (engineering terms and definitions);*

**(H 2).** *Implementation of the developed solution increases the motivation of students for participation in the educational process and leads to higher satisfaction of students;*

**(H 3).** *Students who use mobile platforms and software have better academic achievements, knowledge and learning outcomes than students who do not use mobile systems for learning.*

In this research, the mobile application is developed in cooperation of software engineers and English language teachers to enable more quality teaching and enable the concept of blended and m-learning.

The system was implemented at university, bachelor level in the mechanical engineering study program. Students were divided into two groups (180 students each according to the size of the group defined by accreditation rules in the field of technical and technological sciences).

The lesson that was used to test the system was "Machines, tools and technologies in the field of metal cutting", where one group learned new words and concepts using a mobile application, while the other group learned the given lesson and the associated terms by the classical method of frontal teaching in the classroom.

After the lesson, a short test was conducted, and the achieved results were compared. The level of knowledge of students in both groups and the level of satisfaction with teaching, quality of teaching and learning among students in both groups, and student satisfaction with the application (in the first teaching group using the application) were determined. F and *t*-tests were used in data processing and establishing conclusions.

## 3. Development and Implementation of an Application for Mobile and Smart Platforms Mobil2Eng for English Language Learning

The students at the technical sciences faculties have the subject English as the profession's language, leading to the fact that knowledge of a foreign language provides numerous advantages to graduate engineers. Students of faculties of technical or engineering sciences generally show a high degree of acceptance and use of new technologies while adopting the concept of e-learning, m-learning or blended learning. Therefore, the authors developed an application for mobile and "smart" phones. This application aims to enable faster and more efficient learning of foreign words, (professional) in a new environment where students see certain concepts and objects in their context. The first step was analysis of stakeholders' demands in order to define a high-quality solution that completely fulfils all demands (Table 1). The general objective was to evaluate importance of specific requests for different target groups in order to develop high quality solution. The research covered the first-year students on the Mechanical Engineering study programs and was conducted in 2019. The students were selected from 2 universities (total number 483 students. 67% were male students) also we included 9 English as foreign language teachers from 4 Serbian state universities (teaching English language at mechanical engineering study programs), as well as 4 study program managers and 7 developers (from all four Serbian state universities). Students were questioned as a part of a regular and mandatory evaluation of quality of teaching, learning, and education that universities in Serbia perform at the end of the semester. The requests for the evaluation were listed according to literature [29–32] as well as according to our experience.

**Table 1.** Different requests for systems evaluated by stakeholders (1 the lowest grade, 5 the highest grade).

| | Requests | Students | Teachers | Developers | Institutions |
|---|---|---|---|---|---|
| **Requests for web application** | n | 4.81 | 4.70 | 3.6 | 4.93 |
| | Incorporation of traditional didactic materials in a new context | 3.73 | 4.57 | 3.25 | 4.80 |
| | On-line supervision and monitoring | 4.84 | 4.75 | 4.45 | 4.36 |
| | Flexible control algorithm | 2.81 | 3.99 | 4.31 | 3.19 |
| | Flexible and reliable maintenance system and changes with open and modular architecture | 2.88 | 3.49 | 4.71 | 3.76 |
| | Existence of parameters for quality of service. | 2.17 | 3.31 | 4.78 | 3.52 |
| **Requests for web client** | Multi-platform client software. | 3.29 | 3.11 | 4.63 | 3.29 |
| | Management of changes and distribution of new versions. | 4.03 | 3.98 | 4.81 | 4.17 |
| | Easy installation and maintenance of client software. | 3.54 | 3.44 | 4.61 | 4.92 |
| | Security of client-side application | 3.31 | 3.26 | 3.89 | 3.21 |
| **Requests for course** | Easy access | 4.66 | 4.78 | 4.15 | 4.23 |
| | All exercises and all results should be stored in a database | 4.77 | 4.87 | 4.65 | 4.78 |
| | Storage and download of lectures and audio files | 4.93 | 4.92 | 4.75 | 4.81 |
| | Learning styles need to be considered | 3.33 | 3.81 | 2.45 | 3.77 |

It could be observed that different groups had different focuses, for instance students were interested in user friendly options and possibilities to download material, teachers preferred monitoring and tracking options, and developers were focused on technical characteristics. However, after this analysis, development tools were picked and main functionalities and characteristics developed.

Namely, a mobile application has been developed, which includes the following characteristics:

1. The personalised access and individual login. Three types of system users are predefined. The first type of user is a system administrator who can add new materials and

manage the application. The second type of user is a teacher who can manage the teaching content, direct the course of the teaching process, and have communication with students. The third type of users can, using their identification data, index number and code, access online applications where they learn new words, solve tests, and save selected content. This type is planned so that each student can create their personalised path in learning and the testing itself and receive personalised feedback based on the tests, which would indicate potential weaknesses and strengths in his professional vocabulary.

2. Distribution of teaching materials. The distribution of teaching materials can take place in two ways. The method is designed for students to learn new foreign words, having in mind the specific environment and the objects to which the terms are related. Within the metal processing laboratory, QR codes were set on all machines and more comprehensive tools (Figure 2a). After logging in to the system, the student enters a specific laboratory, approaches the machine and scans the QR code (for example, a milling machine for metal processing). After scanning the code, they received a description of a specific machine in English with photographs (Figure 2b,c), separate professional words and an audio recording where the student can hear the correct pronunciation of the word given (Figure 2d). The student also received related terms, i.e., names of tools (Figure 2e), parts, etc., which are in context with the given initial subject and can create their vocabulary (Figure 2f). In this way, the student learns following his interest, where he sees and connects the concepts in the appropriate context with the possibility of personalisation and choice of his path and pace. In this way, students can use knowledge from other professional subjects and select options that suit them at a given moment (for example, after a lathe, one of the students can scan the code on a completely different machine, and others can deal with parts and tools for the first machine). The system enables scanning of QR code on a given subject, since learning words and languages in specific situations and context is a more favourable option, but the student can also get a printed QR code that will scan and get the appropriate field (in case the classes are conducted entirely in the classroom or if it is a test). Another way of distributing the material is that the teacher directly selects the material that can be sent to a particular group of students, i.e., after logging in to the system, students receive material related to the appropriate work week.

3. Short knowledge tests. After the appropriate teaching block, students can access short tests. Tests are based on the principles of selecting the correct answer or typing one word or one term (Figure 2g). Based on the test results, the student and the teacher receive information on whether the student has mastered and adopted new words and expressions, i.e., which areas he should pay special attention to (Figure 2h). The tests' design includes individualised recommendations that students can receive based on given answers in order to pay particular attention.

4. Tracking student progress. Finally, the teacher has access to the system where he has an overview of all the student accounts, lessons, and QR codes they have chosen, i.e., their tests results. Teachers were allowed to track group (Figure 2i) or individual student progress, lessons used by the student, QR codes he/she loaded, test results, and a list of words that the student entered in his/her dictionary.

5. Constant access to the system and the possibility of individualisation. Students are provided with access to the 24/7 system, where they can search for content and select content according to their priorities, i.e., creating their dictionaries. The nature of the mobile application allows students to access the system and determine the dynamics of their learning and have a platform at their disposal that can be useful in various professional situations.

In the second step, by calculating the Pearson correlations, it was found that a strong positive correlation exists between different requests for mobile application evaluated by students and teachers (0.847) and students and representatives of the institutions (0.725).

Actually, a correlation between the requests of developers and other stakeholders does not exist, and it could be explained by a specific point of view that software developers have on the development of software in this case application for the smart phones and devices (Table 2).

In further analysis, the correlation between specific groups of requests was calculated (Requests for application functionality, Requests for client part of application, and Requests for course). For the first group of requests (Requests for application), there is a strong correlation between students', teachers', and institutional requests (0.926 and 0.811, respectively).

For the second group of requests (Requests for client), there is a strong relation between students and teachers (0.990) as well as students' and institutional requests (0.731). And finally for the third group of requests (Requests for course) there is a strong relation between the requests of all stakeholders. It could be generally concluded that at the "higher" levels (Requests for application, Requests for web client) the differences between stakeholders (developers of the software solution usually have their own point of view) exist, and they are based on their different perspectives towards the stated issue, and at the most basic level (Requests for course), the requests have a strong positive correlation.

**Table 2.** Correlation between different requests for mobile application evaluated by stakeholders.

| | Students | Teachers | Developers | Institutions |
|---|---|---|---|---|
| | Correlation between all requests for application by stakeholders | | | |
| Students | | 0.847 | 0.001 | 0.725 |
| Teachers | 0.847 | | −0.107 | 0.695 |
| Developers | 0.001 | −0.107 | | −0.023 |
| Institutions | 0.725 | 0.695 | −0.023 | |
| | Correlation between "Requests for application" evaluated by stakeholders | | | |
| Students | | 0.926 | −0.529 | 0.811 |
| Teachers | 0.926 | | −0.722 | 0.783 |
| Developers | −0.529 | −0.722 | | −0.773 |
| Institutions | 0.811 | 0.783 | −0.773 | |
| | Correlation between "Requests for web client" evaluated by stakeholders | | | |
| Students | | 0.990 | 0.618 | 0.531 |
| Teachers | 0.990 | | 0.506 | 0.511 |
| Developers | 0.618 | 0.506 | | 0.544 |
| Institutions | 0.531 | 0.511 | 0.544 | |
| | Correlation between "Requests for course" evaluated by stakeholders | | | |
| Students | | 0.998 | 0.991 | 0.901 |
| Teachers | 0.998 | | 0.989 | 0.894 |
| Developers | 0.991 | 0.989 | | 0.948 |
| Institutions | 0.901 | 0.894 | 0.948 | |

The software solution was based, having in mind defined requests, on the technologies necessary for the development of modern mobile applications. The proposed solution has three layers architecture (Figure 1), with typical key elements concerning presentation (employed over mobile devices), application, and data access functionalities (provided by a cloud platform web services), that are physically separated. Dividing components with adaptable relationships into tiers is a proven approach that includes a number of important benefits (and it is a common solution when it comes to web or mobile applications) including overall modularity; maintainability; portability; improved security; mutual independency between technologies and technology stacks used to incorporate various tiers; and separate back-end tier allowing to deploy different databases, with the possibility to scale up, extend by adding multiple web servers, and making decentralized applications.

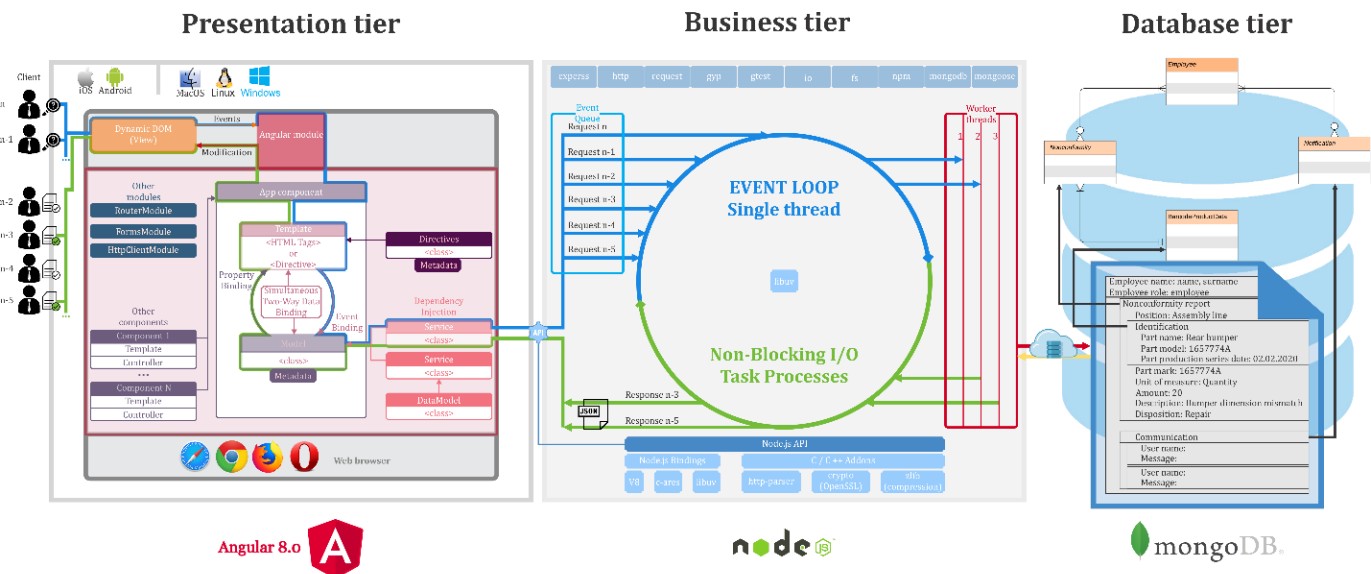

**Figure 1.** Architecture and data flow through adopted technologies.

The presentation tier implies user interface at the applications' top-most level, with the main purpose of handling clients' requests and presenting adequate and understandable results. Development trends mostly include application of JavaScript technologies, JavaScript frameworks or alternative usage of frameworks based on Java. This architecture communication with other tiers is performed through API calls. The application tier is hosted on in-house server. The data access tier provides entry to a data. Additionally, it is possible to easily change data storage technology, i.e., different database management systems (DBMS), moving from one to the other, without significant influence on business logic.

Having in mind the facts stated, it can be observed that a large number of technologies in different layers can be used to implement the software solution. Mean Stack has given a modern approach of web development which runs on every tier of your application. The term MEAN stack refers to a collection of JavaScript based technologies used to develop web applications. MEAN is an acronym for MongoDB, ExpressJS, AngularJS, and Node.js. MEAN presents an open-source web stack that is mainly used to create cloud-hosted applications, since these applications are flexible, scalable, and extensible, making them the perfect candidate for cloud hosting. It is a strong choice for developing cloud native applications because of its scalability and its ability to manage concurrent users, so we based our solution on these technologies (Figure 1).

If we go back to the requirements, it could be clearly concluded that all technical requirements were fulfilled by making appropriate framework selection and course specific requirements for teaching and learning were fulfilled with appropriate selection of modules and functionalities. The request such as incorporation of traditional didactic materials in a new context was enabled through scanning of QR code in laboratory or real working environment. Moreover, requirement of consideration of different learning styles is supported by the possibility of creating different learning paths. Some requests (Multi-platform client software, Management of changes and distribution of new versions, Flexible and reliable maintenance system and changes with open and modular architecture, Security of client-side application, Easy understanding and usage, Easy installation and maintenance of client software) were supported with a technical solution based on MEAN stack. Finally, solution employs cloud storage system for storage and download of lectures and audio files as well as results of testing.

The developed application is presented in Figure 2, with a selection of characteristic screens that correspond to the basic functionalities listed above. Of course, the presented application for mobile systems has all the characteristics that have such features: open-

ness, mobility, 24/7 access, intuitive user interface, and the ability to expand and modify the purpose.

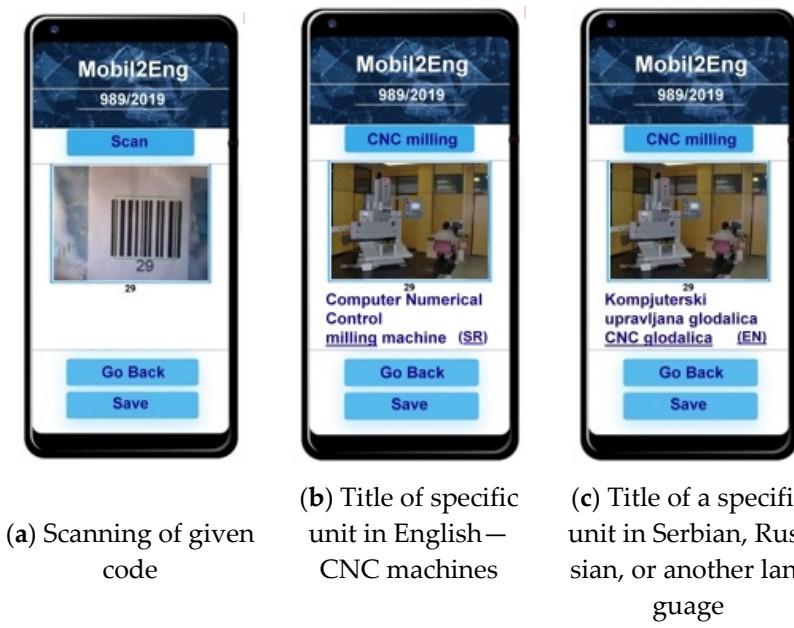 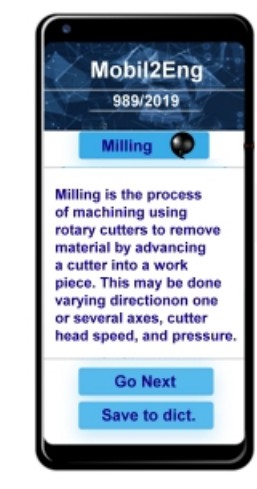 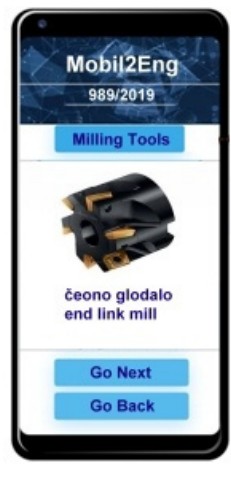

(**a**) Scanning of given code

(**b**) Title of specific unit in English—CNC machines

(**c**) Title of a specific unit in Serbian, Russian, or another language

(**d**) Description of the concept and possibility of playing an audio file

(**e**) Continuation of the lesson—description of the tool

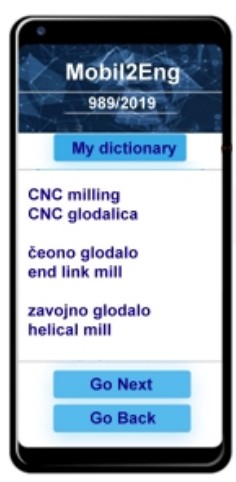 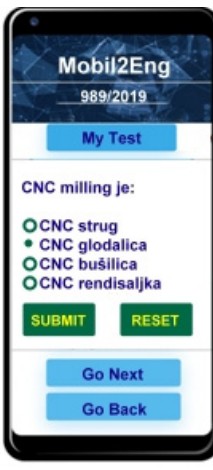 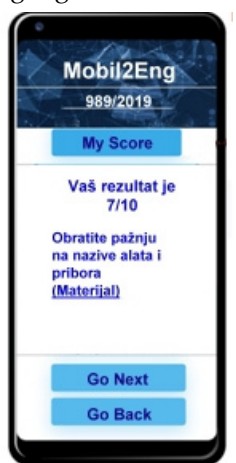 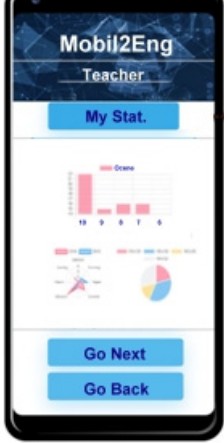

(**f**) My Dictionary option

(**g**) My Test option

(**h**) Test result and recommendation

(**i**) Evaluation statistics

**Figure 2.** Screenshots of characteristic options of Mobil2Eng application.

In [33] the following applications were analysed as mobile apps used for Tertiary Level Learners (University level and adult learners): Sounds Right, WordBook XL—English Dictionary & Thesaurus, Speech Tutor, English Podcast for Learners, Voxy, English Listening and Speaking, Exam Vocabulary Builder, Learn English with busuu.com!, Sentence Builder and Learn English, Speak English. Most of these solutions have options for vocabulary acquisition, pronunciation, audio files, illustrative sentences, Grammar learning, and video clips. The presented solution Mobil2Eng has clear advantages in both technical and educational senses. In the technical part, Mobil2Eng is developed using MEAN stack; the main data base is on the cloud, so all the client part of software is easy to download and does not demand a specific environment. In addition, all data on the cloud could be used as an integral dictionary or resource base. Mobil2Eng enables users to read QR codes and to get information, related words, and explanation connected to specific objects. For instance, when users scan the QR code on milling machine they will get definition of specific term, related terms, and conditions as well as additional audio and video material. Moreover,

Mobil2Eng enables users and instructors (teachers) to keep track of achievements and to define individual learning and self-regulated learning (for instance, learning path for mechanical engineering, industrial engineering, metal forming, etc.) Mobil2Eng has full support and analytics for the instructor, so it is much more suitable for usage in classrooms.

The application was tested from two aspects to research the set hypotheses, i.e., to assess student satisfaction and the efficiency of the system.

## 4. Results of the Implementation of the Mobil2Eng Application

The application was tested with students from the freshmen year at Faculty of Engineering, University of Kragujevac, Serbia. Students had course English language 1 during the first semester. All students were divided (by random selection made by faculty) in two lecturing groups. Both groups had equal number of students 180 each (74% were male students). During the semester, the groups that used the system were reduced to 164 students. The questions were defined by Quality Control Commission as a part of regular quality control procedure. After using the application for the subject English Language 1, the goal was to evaluate students' satisfaction as users of the application and determine whether the application of the developed solution gives some positive results in the teaching process, primarily in learning new words and technical expressions.

Table 3 shows the results of a survey given to students who used the application. The aim was to investigate the characteristics of the developed application (method of use, intuitiveness, etc.), the contribution of the application in increasing the multimedia character of the course and improvement in teaching, and also the assessment of students whether this approach allows them a higher level of satisfaction and individualisation of learning creating individual learning paths. The results presented in Table 3 prove that students are satisfied with the introduced application and believe that it improves teaching.

**Table 3.** Students' satisfaction with the usage of the system in the learning process.

| No | Question | The Grade Ranges from 1 to 5, with One Being the Lowest and 5 Being the Highest Grade (a Total of 164 Students Were Surveyed in the Teaching Group That Used the System) | | | | |
|----|----------|----|----|----|----|----|
| 1 | System is user friendly and easy to use | 2% | 5% | 10% | 37% | 46% |
| 2 | A learning outcome is clear and educational goals are well presented in the system | 5% | 10% | 17% | 35% | 33% |
| 3 | System is reliable, and intuitive | 6% | 10% | 19% | 36% | 29% |
| 4 | System facilitates better learning and has number of services | 0% | 14% | 15% | 26% | 45% |
| 5 | Digital "e" character of the course is increased | 0% | 0% | 10% | 23% | 67% |
| 6 | System presents clear improvement | 0% | 2% | 7% | 22% | 69% |
| 7 | System is stimulative and interesting | 0% | 6% | 20% | 36% | 38% |
| 8 | System enables easier vocabulary acquisition | 2% | 12% | 22% | 30% | 34% |
| 9 | System enables individual learning paths | 3% | 7% | 19% | 42% | 29% |
| 10 | Evaluation of total satisfaction with the system | 2% | 9% | 18% | 22% | 49% |

Different research reported that students are generally satisfied with usage of mobile and smart platforms in education [34]. The research [35] presented results from a survey of 19 students where students evaluated that mobile learning systems are easy to use and understand 3.0, easy for distribution of material 3.42, and convenient for discussion 3.79 as well as satisfaction with learning tool 3.74. These data generally correspond to data we obtained in our analysis. According to our research, it is clear that students are accepting new technologies which corresponds with [36] that reports that students at online universities have started to accept mobile technology as a new learning tool; consequently, its acceptance has influenced their learning achievement both directly and indirectly. When it comes to the learning of the English language, our research indicates

an increasing motivation of students that have been using mobile platform for learning which corresponds to similar research [37]. One point where the presented application has clear advantage comparing students' satisfaction is that students evaluate existence of individual learning paths (enabled by system) very high (total 71% of 164 evaluated this feature with 5 or 4, 29% with a maximal grade). Moreover, this system has higher grade in the field of being user friendly to end user.

For the aspect of learning, the system was tested on one group of 180 students (according to the norms for the technical, technological field) using the "Machines, tools and technologies in the field of metal cutting course". Students studied the lesson with the system's help, visited the laboratory, used the system for learning words, and monitored their lesson learning process (where the first part of the teaching was in the classroom, and the second was done in groups using the system). The second group (control) of 180 students listened to the same lesson in the classroom using classical teaching methods. After the teaching unit, students were given a test to assess their knowledge, emphasising the vocabulary adopted after a specific teaching unit.

Table 4 shows the scores after the short test that both groups had. The mean score in the first group was 8.44 and in the control group 7.93, with a variance of 1.41 and 1.89, respectively.

F-test and *t*-test were performed. The critical value of the parameter for the F-test is 0.66, and it is higher than the calculated F value of 0.74, so the assumption that the standard deviations in both groups are low is maintained (Table 4 F-test). The *t*-test was then carried out to determine if there was a difference between the test and control groups (Table 5 *t*-test).

**Table 4.** Test score in the examined group and control group (values are given—number of students who received a particular grade).

|  | I Experimental Group | II Control Group |
|---|---|---|
| Grade 10 | 37 | 25 |
| Grade 9 | 46 | 36 |
| Grade 8 | 49 | 43 |
| Grade 7 | 23 | 37 |
| Grade 6 | 8 | 16 |
| Grade 5 | 2 | 8 |

**Table 5.** Results of F-test and *t*-test.

| | F-Test | |
|---|---|---|
| | **Group 1** | **Group 2** |
| Mean | 8.454545 | 7.951515 |
| Variance | 1.420177 | 1.912269 |
| Observations | 165 | 165 |
| df | 164 | 164 |
| F | 0.742666 | |
| P(F ≤ f) one-tail | 0.028809 | |
| F Critical one-tail | 0.772882 | |
| | ***t*-Test** | |
| | **Group 1** | **Group 2** |
| Mean | 8.454545 | 7.951515 |
| Variance | 1.420177 | 1.912269 |
| Observations | 165 | 165 |
| df | 328 | |
| t Stat | 3.539603 | |
| P(T ≤ t) one-tail | 0.000229 | |
| t Critical one-tail | 1.649512 | |
| P(T ≤ t) two-tail | 0.000459 | |
| t Critical two-tail | 1.967223 | |

Given that t Stat = 3.539603 and is more significant than t Critical two-tail = 1.967223, the conclusion is that there is a significant difference in the study group results (which used the mobile application) and the control group. The test has been repeated three times, and the differences in the average score were 0.47 and 0.63. In all three cases, students who studied with the help of the application achieved better results. According to above said, it has been proven that students in the group that had a mobile application at their disposal in the e-learning process achieve better results in acquiring professional vocabulary. This result generally corresponds to the other research showing that usage of m-learning contributes to improved learning success [38]. The research conducted on experimental and control group (with 50, 25 each) students [39] also reported that students that used mobile solution for learning English achieve better results. We presented research on learning English as foreign language for professionals in the field of engineering sciences, and these findings also reported that students that used mobile devices and software solutions achieve better results, but we also presented distribution of grades which indicates that the number of students with lower grades significantly decreases.

## 5. Discussion and Conclusions

The application of various tools and concepts in e-learning has not been new for a long time. In contrast, at the level of the university, academic education, a large number of courses have significant support in e-learning tools but also m-learning, whether it is just the distribution of electronic content, communication, or more advanced methods such as the use of e-classrooms or systems for electronic management of teaching content. It is difficult to say that there are pure examples of e-learning, but different forms of mixed forms of teaching, learning, and education are becoming increasingly important.

One of the courses that are present in almost all engineering study programs is the teaching of a foreign language (mainly as the language of the profession), which by its nature and place in the curriculum of engineering studies is set as a general educational subject and often does not receive significant attention. It is also important to note that according to the rules for accreditation in the technical-technological field, the teaching group's size is 180 students, and for auditory exercises, that number is 60. According to the mentioned numbers, the content is supported by appropriate technical means. According to the above numbers, it is not easy to organise classes using any other teaching method than the most classic frontal approach or possible visual presentation of the content using appropriate technical means.

Mobile learning is becoming increasingly important for foreign language learning. Its key benefits are as follows: the enhancement of the learner's cognitive capacity, the learner's motivation to study in both formal and informal settings, the learner's autonomy and confidence, and the promotion of personalized learning, helping low-achieving students to reach their study goals [40]. We have developed an application for mobile systems Mobil2Eng, which has the task of modernising the teaching process in English 1 in engineering studies. The application was created as a solution for mobile phones, using modern development environments. They allow the application to be easy to use, to be independent of the type of mobile device, and to achieve a range of functionalities that give students a more interactive, intuitive, and individualised approach to mastering the technical profession's words and expressions. Students get related terms, pronunciation, descriptions in Serbian and English, in addition to the opportunity to test their knowledge and create their dictionaries and learning focuses.

The results of this paper indicate that using applications for mobile systems in the concept of "blended" learning has a noticeable positive effect, in terms of encouraging better understanding, learning, and better academic achievement of students. While compared to traditional receptive teaching, initial research shows positive shift in motivation and satisfaction, it is still necessary to monitor the system in the classroom for a long time.

The system itself was tested on students in the course English language, and based on the results presented in Tables 1 and 2, the following can be concluded:

**(H 1).** *It is possible to develop software solutions for mobile platforms that will lead to more interactive teaching, and better mastering of materials, especially mastering engineering vocabulary*, has been proven since the application was created and based on knowledge testing in the test and control group. It was found that students who used mobile application for learning have better grades (on average higher by 0.47, 0.51, and 0.63, on repeated surveys). Moreover, a total of 64% (Table 1) of students expressed a very positive attitude regarding the fact that the mobile system makes it easier to master engineering vocabulary.

**(H 2).** *The application of advanced solutions increases students' motivation to learn and leads to higher student satisfaction with the teaching process*, is substantiated by survey data presented in Table 3, where 49% of students rated their satisfaction with the maximum grade.

**(H 3).** *Better academic achievements are achieved by students who use modern mobile systems to support learning as compared to the classical approach to teaching*, which is proven by testing on the examined and control group and proven by the application of F and t-test. The first group had the opportunity to use the application, and the second did not (Table 4).

Based on the above mentioned, it can be stated that the research goal is achieved and that the initial hypotheses are proven. In addition, students had the opportunity to submit their comments in free form during the survey. The most frequent comments were that the system is exciting and motivates them for further work and learning (since one concept is straightforward to use by connecting links to another) and that it is beneficial for them to create the technical dictionary that allows them to store and organise words. and the expressions they consider most important. It is clear that the use of such an application for mobile devices is useful in teaching a foreign language in academic studies and that it represents another contribution to the creation of mixed forms of teaching and learning and a step towards m-learning.

The presented software solution has a clear advantage compared to other solutions, mainly in distribution of material as well as on innovative acquisition system based on QR codes. In addition, students highly evaluated the possibility to create individual learning paths using this system, a feature which is another advantage of the presented solution and an advantage when comparing with other systems. From the technical point of view, the solution is based on MEAN stack, taking a number of advantages from this framework which enable portability, scalability, flexibility, and cloud-based functionality.

It has been proven that the application of such solutions has a motivating effect on students and enables them to achieve better results. Of course, the system has limitations in terms of the scope of words and terms entered and the scope of functionality, so one of the directions for further work will certainly be the technical improvement of the application and expanding the database of words, terms, and lessons. Moreover, the application can change the content and can be used for other areas, not only English for the technical profession. Apart from this, further work directions can include integrating this teaching solution with content management systems such as Moodle, for example. The system can also be improved to support other segments of language learning, such as grammar. In any case, the presented system has significant advantages and is open for further improvements.

**Author Contributions:** Conceptualization, S.S. and E.K.; methodology, E.K. and S.S.; software, S.S.; validation, E.K. All authors have read and agreed to the published version of the manuscript.

**Funding:** This research was funded by the Ministry of Science and Higher Education of the Russian Federation, grant number 075-15-2020-934.

**Institutional Review Board Statement:** Not applicable.

**Informed Consent Statement:** Not applicable.

**Data Availability Statement:** Not applicable.

**Acknowledgments:** The research is partially funded by the Ministry of Science and Higher Education of the Russian Federation as part of World-class Research Center program: Advanced Digital Technologies (contract No. 075-15-2020-934 dated 17 November 2020).

**Conflicts of Interest:** The authors declare no conflict of interest.

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
