# Peer review of "Digitalisation of Teaching and Learning as a Tool for Increasing Students’ Satisfaction and Educational Efficiency: Using Smart Platforms in EFL"

_sustainability, doi:10.3390/su13094892_

Round 1

Reviewer 1 Report

Dear Authors,

Your work was interesting to read; thank you for sharing your research within the scientific community. Please correct a few typos in the text, and check the editing of some references.

I wish you good luck with the next steps.

Author Response

We performed spelling checker and we corrected typos. Also we checked and corrected references.

Sincerely,

authors

Reviewer 2 Report

The article is written in the appropriate style. All parts of it are balanced and corresponds its structure.

Author Response

Thank you very much for your review and comments.

Sincerely,

Authors

Reviewer 3 Report

Dear authors, as a result of reading your article, I have some observations and recommendations, as follows: 
-    considering that two researches were carried out, one regarding identification of the requirements of the potential users and another regarding the opinion of those who used the application, I consider that the methodology of both researches should be properly presented (specifying the purpose, objectives, construction of the sample, period and method of data collection, hypotheses), and the results should be correlated, in order to clearly indicate the extent to which those requirements identified in the preliminary phase have been met by the mobile application;
-    the design stages of the application and its basic technical characteristics should be presented. The architecture of the proposed solution for the mobile application should be underlined in more detail, with reference to components such as algorithms for optimizing data management, security, and infrastructure monitoring;
-    also, the particularities of the application should be highlighted. It must be understood what differentiates this application from other m-learning applications, but also what are its attributes that make it relevant and useful especially for students in the technical/engineering field;
-    the results of the research are not related to other studies and articles that presented the results of other researches regarding the requirements, respectively the degree of satisfaction of the users of such applications, in order to highlight the authors' contribution in this respect. Also in this sense, I mention that the bibliography is penurious, confirming the weak reference to scientific sources of documentation.

Author Response

Dear reviewer,

Thank you for your observations and comments. We did our best to provide all explanation and additional information and to meet your requests. In the text below we tried to address all issues and also we incorporate changes in the manuscript. All changes in the manuscript are marked with red color.

Best regards,

Authors

Dear authors, as a result of reading your article, I have some observations and recommendations, as follows: 

-    considering that two researches were carried out, one regarding identification of the requirements of the potential users and another regarding the opinion of those who used the application, I consider that the methodology of both researches should be properly presented (specifying the purpose, objectives, construction of the sample, period and method of data collection, hypotheses), and the results should be correlated, in order to clearly indicate the extent to which those requirements identified in the preliminary phase have been met by the mobile application;

The general objective was to evaluate importance of specific requests for different target groups in order to develop high quality solution. The research covered the first year students on the Mechanical Engineering study programs and was conducted in 2019. The students were selected from 2 Universities (total number 483 students. 67% were male students) also we included 9 English as foreign language teachers from 4 Serbian state Universities (teaching English language at Mechanical engineering study programs), as well as 4 study program managers and 7 developers (from all four Serbian state Universities). Students were questioned as a part of a regular and mandatory evaluation of quality of teaching, learning and education that Universities in Serbia perform at the end of the semester. The requests for the evaluation were listed according to literature [29, 30, 31, 32] as well as according to our experience.

Table 2. Correlation between different requests for mobile application evaluated by stakeholders

(Table is presented in attached file)

In the second step by calculating the Pearson correlations, it was found that a strong positive correlation exists between different requests for mobile application evaluated by students and teachers (0.847), students and representatives of the institutions (0.725).Actually a correlation between the requests of developers and other stakeholders does not exist and it could be explained by a specific point of view that software developers have on the development of software in this case application for the smart phones and devices(Table 2).

In further analysis the correlation between specific groups of requests was calculated (Requests for application functionality, Requests for client part of application and Requests for course). For the first group of requests (requests for application) there is a strong correlation between students’, teachers’ and institutional requests (0,926 and 0,811 retrospectively).

For the second group of requests (Requests for client), there is a strong relation between students and teachers (0,990) as well as students’ and institutional requests (0.731). And finally for the third group of requests (Requests for course) there is a strong relation between the requests of all stakeholders. It could be generally concluded that at the “higher” levels (Requests for application, Requests for web client) the differences between stakeholders (developers of the software solution usually have their own point of view) exist and they are based on their different perspectives towards the stated issue, and at the most basic level (Requests for course) the requests have a strong positive correlation.

The application was tested with students from the freshmen year at Faculty of Engineering, University of Kragujevac Serbia. Students had course English language 1 during the first semester. All students were divided (by random selection made by faculty) in two lecturing groups. Both groups had equal number of students 180 each. During the semester the groups that used system reduced to 164 students.

---------------------------------------------------------

If we go back to the requirements it could be clearly concluded that all technical requirements were fulfilled by making appropriate framework selection and course specific requirements for teaching and learning were fulfilled with appropriate selection of modules and functionalities. The request such as incorporation of traditional didactic materials in a new context was enabled through scanning of QR code in laboratory or real working environment. Also requirement of consideration of different learning styles is supported by possibility of creating of different learning paths. Some requests (Multi-platform client software, Management of changes and distribution of new versions, Flexible and reliable maintenance system and changes with open and modular architecture, Security of client-side application, Easy understanding and usage, Easy installation and maintenance of client software) were supported with technical solution based on MEAN stack. Finally solution employees cloud storage system for storage and download of lectures and audio files as well as results of testing).

  1. de la Iglesia, D. G., Andersson, J., & Milrad, M. (2010, November). Enhancing mobile learning activities by the use of mobile virtual devices--some design and implementation issues. In 2010 International Conference on Intelligent Networking and Collaborative Systems (pp. 137-144). IEEE.

30.Mcconatha, D., Praul, M., & Lynch, M. J. (2008). Mobile learning in higher education: An empirical assessment of a new educational tool. Turkish Online Journal of Educational Technology-TOJET, 7(3), 15-21.

  1. Sarrab, M., Elbasir, M., & Alnaeli, S. (2016). Towards a quality model of technical aspects for mobile learning services: An empirical investigation. Computers in Human Behavior, 55, 100-112.
  2. Stefanovic, M., Matijevic, M., Cvijetkovic, V., & Simic, V. (2010). Web‐based laboratory for engineering education. Computer Applications in Engineering Education, 18(3), 526-536.

-    the design stages of the application and its basic technical characteristics should be presented. The architecture of the proposed solution for the mobile application should be underlined in more detail, with reference to components such as algorithms for optimizing data management, security, and infrastructure monitoring; 

The software solution was based, having in mind defined requests, on the technologies necessary for the development of modern mobile applications. The proposed solution has three layers architecture (Figure 1), with typical key elements concerning presentation (employed over mobile devices), application and data access functionalities (provided by a cloud platform web services), that are physically separated. Dividing components with adaptable relationships into a tiers is proven approach that includes number of important benefits (and it is common solution when it comes to web or mobile applications) including overall modularity; maintainability; portability; improved security; mutual independency between technologies and technology stacks used to incorporate various tiers; and separate back-end tier allowing to deploy different databases, with the possibility to scale up, extend by adding multiple web servers, and making  decentralized applications.

The presentation tier implies user interface at the applications' top-most level, with the main purpose to handle clients' requests and to present adequate and understandable results. Development trends mostly include application of JavaScript technologies, JavaScript frameworks or alternative usage of frameworks based on Java. This architecture communication with other tiers is performed through API calls. The application tier is hosted on in-house server. The data access tier provides entry to a data. It makes possible to easily change different database management systems (DBMS), without significant influence on business logic.

Having in mind the facts stated, it can be observed that a large number of technologies in different layers can be used to implement the software solution. Mean Stack has given a modern approach of web development which runs on every tier of your application. The term MEAN stack refers to a collection of JavaScript based technologies used to develop web applications. MEAN is an acronym for MongoDB, ExpressJS, AngularJS and Node.js. MEAN presents an open-source web stack that is mainly used to create cloud-hosted applications, since these applications are flexible, scalable, and extensible, making them the perfect candidate for cloud hosting. It’s a strong choice for developing cloud native applications because of its scalability and its ability to manage concurrent users so we based our solution on these technologies (Figure 1). Technology selection can be considered as an essential aspect, as it affects the cost, performance and possible functionalities.

Figure 1. Architecture and data flow through adopted technologies

-    also, the particularities of the application should be highlighted. It must be understood what differentiates this application from other m-learning applications, but also what are its attributes that make it relevant and useful especially for students in the technical / engineering field;

In [33] the following applications were analyzed as mobile apps used for Tertiary Level Learners (University level and adult learners): Sounds Right, WordBook XL – English Dictionary & Thesaurus, Speech Tutor, English Podcast for Learners, Voxy, English Listening and Speaking, Exam Vocabulary Builder, Learn English with busuu.com!, Sentence Builder and Learn English, Speak English. The most of these solutions have options for vocabulary acquisition, pronunciations, audio files, illustrative sentences, Grammar learning, video clips. The presented solution Mobil2Eng has clear advantages in both technical and educational sense. In technical part Mobil2Eng is developed using MEAN stack, the main data base is on the cloud so all the client part of software is easy to download and does not demand specific environment. In addition all data on the cloud are could be used as integral dictionary or resource base. Mobil2Eng enables users to read QR codes and to get information, related words and explanation connected to specific object. For instance we user scan QR code on milling machine he /she will get definition of specific term, related terms and conditions as well as additional audio and video material. Also Mobil2Eng enables users and instructors (teachers) to keep track about achievements and to define individual learning and self regulated learning (for instance learning path for mechanical engineering, industrial engineering, metal forming…). Mobil2Eng has full support and analytics for instructor so is much more suitable for usage in classrooms.

  1. Gangaiamaran, R., & Pasupathi, M. (2017). Review on use of mobile apps for language learning. International Journal of Applied Engineering Research, 12(21), 11242-11251.

  -   the results of the research are not related to other studies and articles that presented the results of other researches regarding the requirements, respectively the degree of satisfaction of the users of such applications, in order to highlight the authors' contribution in this respect. Also in this sense, I mention that the bibliography is penurious, confirming the weak reference to scientific sources of documentation.

Different researches reported that students are generally satisfied with usage of mobile and smart platforms in education [34]. The research [35] presented results from survey of 19 students where students evaluated that mobile learning systems are easy to use and understand 3.0, easy for distribution of material 3.42 and convenient for discus-sion 3.79 as well as satisfaction with learning tool 3.74. These data generally corresponds to data we obtained in our analysis. According to our research it is clear that students are accepting new technologies which corresponds with [36] that reports that students at online universities have started to accept mobile technology as a new learning tool; con-sequently, its acceptance has influenced their learning achievement both directly and in-directly. When it comes to learning of English language our research indicates on increasing motivation of students that have been using mobile platform for learning which corresponds to similar researches [37]. One point where presented application has clear advantage comparing students’ satisfaction is that students evaluate existence if individual learning paths (enabled by system) very high (total 71% of 164 evaluated this feature with 5 or 4, 29% with a maximal grade). Also this system has higher grade in the field of being user friendly to end user.

-------------------------------------------------------------------

This result generally corresponds to the other research showing that usage of m-learning contributes to improved learning success [38]. The research conducted on experimental and control group (with 50, 25 each) students [39] also reported that students that used mobile solution in learning of English language achieve better results. We presented research of learning English as foreign language for professionals in the field of engineering sciences and these findings also reported that students that used mobile devices and software solution achieve better results, but we also presented distribution of grades which indicates that number of students with lower grades significantly decreases.

-------------------------------------------------------------------

Mobile learning is becoming increasingly important for foreign language learning. Its key benefits are as follows: the enhancement of the learner’s cognitive capacity, the learner’s motivation to study in both formal and informal settings, the learner’s autonomy and confidence, as well as the promotion of personalized learning, helping low-achieving students to reach their study goals [39].

-------------------------------------------------------------------

The presented software solution has clear advantage comparing to other solution mainly in distribution of material as well as on innovative acquisition system based on QR codes. In addition students highly evaluated possibility to create individual learning paths using this system, the feature that is another advantage of presented solution and comparatively advantage comparing other systems. Besides this software solution has option for tracking students’ success as well as analytics and provides better set of functionalities for teachers and instructors comparing other available solutions. From technical point of view the solution is based on MEAN stack taking the number of advantages from this framework enabling portability, scalability, flexibility and cloud based functionalities.

34. Sulaiman, A., & Dashti, A. (2018). Students’ satisfaction and factors in using Mobile learning among college students in Kuwait. EURASIA Journal of Mathematics, Science and Technology Education, 14(7), 3181-3189.

35. Motiwalla, L. F. (2007). Mobile learning: A framework and evaluation. Computers & education, 49(3), 581-596.

36. Shin, W. S., & Kang, M. (2015). The use of a mobile learning management system at an online university and its effect on learning satisfaction and achievement. International Review of Research in Open and Distributed Learning, 16(3), 110-130.

37. Huang, C. S., Yang, S. J., Chiang, T. H., & Su, A. Y. (2016). Effects of situated mobile learning approach on learning motivation and performance of EFL students. Journal of Educational Technology & Society, 19(1), 263-276.

38. Cavus, N., & Ibrahim, D. (2009). m‐Learning: An experiment in using SMS to support learning new English language words. British journal of educational technology, 40(1), 78-91.

39. Elfeky, A. I. M., & Masadeh, T. S. Y. (2016). The Effect of Mobile Learning on Students' Achievement and Conversational Skills. International Journal of higher education, 5(3), 20-31.

40. Kacetl, J., & Klímová, B. (2019). Use of smartphone applications in english language learning—A challenge for foreign language education. Education Sciences, 9(3), 179.

-------------------------------------------------------------------

Finally we added 11 new references in the reference list.

Reviewer 4 Report

The submission focuses on a very timely issue, the teaching of ESL with mobile applications. The authors concentrate on promoting foreign language proficiency among engineering students with special attention to professional terms and expressions. While the authors recognise that language training in case of engineering programs does not enjoy priority, they provide a justification for teaching English language skills. They constructed three hypotheses which are substantiated by questionnaire-based surveys and the results of the attendant statistical analysis are summarized in tables.  The article uses a wide variety of professional research literature and the authors demonstrate their familiarity with the latest trends of electronics supported learning.

Author Response

Dear reviewer,

Thank you very much for your comments and suggestions. We provided proof reading and we made all corrections that you pointed on.

Sincerelly,

Authors

Round 2

Reviewer 3 Report

Dear authors,

Thank you for the opportunity to read and review your article.

Given that you have considered the suggestions in the previous review report, I appreciate that the article is now ready for publication.